# Nutritional Status of Patients with Neoplasms Undergoing Ambulatory Chemotherapy and Associated Factors

**DOI:** 10.3390/nu17010168

**Published:** 2025-01-02

**Authors:** Luiz Claudio Barreto Silva Neto, Oscar Geovanny Enriquez-Martinez, Wesley Rocha Grippa, Julia Anhoque Cavalcanti Marcarini, Thayná Borges Santos, Nina Bruna de Souza Mawandji, Karoline Neumann Gomes, Sara Isabel Pimentel de Carvalho Schuab, Etreo Junior Carneiro da Silva Minarini, Karolini Zuqui Nunes, Andressa Bolsoni-Lopes, Luís Carlos Lopes-Júnior

**Affiliations:** 1Graduate Program in Public Health, Universidade Federal do Espírito Santo, Vitória 29075-910, ES, Brazil; 2Graduate Program in Nutrition and Health, Universidade Federal do Espírito Santo, Vitória 29075-910, ES, Brazilkarol-zuqui@hotmail.com (K.Z.N.);; 3Graduate Program in Physiological Sciences, Federal University of Espírito Santo, Vitória 29075-910, ES, Brazil

**Keywords:** cancer, nutritional status, chemotherapy, associated factors, cancer care, oncology

## Abstract

Background/Objectives: Cancer, a leading cause of mortality globally and in Brazil, is influenced by environmental and behavioral factors, often linked to nutritional deficiencies such as low body mass index and muscle wasting, exacerbating prognostic outcomes and mortality rates. Timely nutritional interventions during chemotherapy are pivotal, necessitating continuous nutritional assessment for effective patient care management. This study aimed to assess the nutritional status of non-metastatic cancer patients undergoing chemotherapy and identify factors influencing their nutritional status. Patient evaluation involved sociodemographic data, clinical profiles, anthropometric measurements, blood biochemical analyses, and nutritional status classification employing the Patient-Generated Subjective Global Assessment (PG-SGA) criteria. Statistical analysis was performed using R software. Results: Suspected malnutrition was identified in 5.81% of patients, with a significant association observed with gender, indicating a higher prevalence among men. Cancer stages II and III, along with a positive family history, correlated with heightened risk of malnutrition. Patients with suspected malnutrition exhibited older age, lower weight, body mass index (BMI), and reduced circumferences, underscoring the necessity of comprehensive nutritional assessment for optimized patient management during treatment. Conclusions: This study underscores a notable prevalence of malnutrition, particularly among patients with lower weight and BMI, affirming the reliability of PG-SGA criteria.

## 1. Introduction

Neoplasms represent the second leading cause of global mortality, with an escalating incidence spanning all continents, irrespective of their level of human development. This rise is intricately linked to the exacerbation of factors associated with globalization and economic growth [1,2]. In Brazil, projections indicate an annual estimate of 704,000 new cancer cases for the three-year period spanning 2023 to 2025, excluding non-melanoma types [3].

Adverse shifts in lifestyle, behavior, and environmental factors have markedly contributed to the surge in cancer incidence and mortality. Structural transformations affecting mobility, leisure activities, dietary patterns, and exposure to environmental pollutants have played a pivotal role in this regard [3].

The growing global challenge posed by cancer has driven the search for innovative strategies to enhance treatment efficacy and improve clinical outcomes for patients [4]. In this context, immunonutrition has emerged as a promising approach, with the potential to optimize therapeutic response and promote a better quality of life during treatment [5].

However, studies have highlighted significant gaps in clinical practice [6,7], showing that many still face difficulties in implementing proper nutritional management and assessment due to a lack of training and financial resources [6,7], as well as the need for more effective protocols and tools to better measure nutritional status [6,7].

Assessing and understanding the nutritional status in cancer patients is a challenge, as with the start of treatment, unintentional weight loss is associated with a less favorable prognosis [8]. The Subjective Global Assessment (SGA) is a clinical tool used to evaluate patients’ nutritional status, drawing upon their medical history and physical examination findings [9,10], recommended by ESPEN for nutritional status assessment [11]. Its primary objective is to identify individuals with inadequate nutritional intake and absorption, thus necessitating nutritional intervention [9,10].

One of the factors that directly impacts nutritional status in the antineoplastic treatment is that chemotherapy treatment heightens the risk of malnutrition and weight loss, necessitating thorough nutritional assessment to preserve body weight and enhance patient prognosis [10,12]. Ongoing monitoring is indispensable to safeguard the nutritional status of patients commencing or undergoing chemotherapy regimens [10,13]. A comprehensive nutritional evaluation during tumor therapy is paramount, ideally based on meticulous analysis of patients’ status. Studies underscore nutrient deficiency as a prognostic risk factor in cancer patients, with nutritional status correlating with diminished survival and prolonged hospitalization [14,15].

Given the above, the imperative of timely and adequate nutritional assessment for outpatient chemotherapy patients becomes evident. Such assessment should precede treatment initiation, aiming to promptly identify nutritional risk factors. Moreover, continual evaluation during therapy is essential to optimize patient outcomes [10].

The existing literature lacks detailed nutritional assessments of patients and identification of associated factors, which could inform superior nutritional care protocols [16,17]. Thus, the objective of this study is to assess the nutritional status of non-metastatic cancer patients undergoing chemotherapy and delineate factors linked to nutritional status.

## 2. Materials and Methods

### 2.1. Study Design 

A cross-sectional study was conducted at the High Complexity Oncology Care Center (AFECC—Santa Rita de Cássia Hospital—HSRC) situated in Vitória, Espírito Santo, in the southeast region of Brazil.

### 2.2. Ethical Aspects

The study received approval from the Research Ethics Committee of the Health Sciences Center (CEP/CCS/UFES), with CAAE Number: 32362420.6.0000.5060. All participants were invited to partake and provided written consent using a free and informed consent form, ensuring confidentiality of their data.

### 2.3. Study Sample

The sample size calculation considered the patient population from the service where recruitment occurred, as well as previous studies conducted at the hospital [18,19]. The following formula was utilized for sample calculation: *n* = *N*.*Z*^2^.*p*.(1 − *p*)/*Z*^2^.*p*.(1 − *p*) + *e*^2^.*N* − 1 (where *n*: calculated sample; *N*: population; *Z*: normal variable; *p*: real probability of the event; *e*: sampling error) [18].

Considering the patient population diagnosed (only new cases) at AFECC-Santa Rita de Cássia Hospital in 2022 (*n* = 3513) without the bias of the COVID-19 pandemic; and taking into account the total number of new cases in the state of Espírito Santo in 2022 (National Cancer Institute [20] *n* = 10,880); and assuming at least 20% of the total new AFECC-HSRC cases meeting all inclusion/exclusion criteria (703 patients—representing 6.45% of the incidence of new cases in Espírito Santo for 2021–2022); with α set at 5% (sampling error), confidence level at 95%, and test power at 80% (β = 0.20), the sample size for this research was determined to be 84 patients.

### 2.4. Eligibility

The inclusion criteria were (a) age over 18 years; (b) anatomopathological diagnosis of cancer, according to the International Classification of Diseases (ICD-10), in stages I, II, III, or IV; and (c) patients (new cases only) who underwent any type of antineoplastic treatment. Exclusion criteria were (a) patients with a previous history of cancer treatment (chemotherapy or radiotherapy); and (b) patients receiving exclusive palliative care; (c) patients with multiple primary tumors.

### 2.5. Assessment of Nutritional Status

Nutritional status was evaluated utilizing the Patient-Generated Subjective Global Assessment (PG-SGA), a globally recognized tool validated for Brazilian Portuguese [21]. This assessment comprises two stages: firstly, patients self-assess factors such as involuntary weight loss, alterations in food intake, and symptomatic manifestations impacting nutrition. Subsequently, factors associated with metabolic stress, physical depletion, and other health conditions are evaluated. Nutritional status is then classified as “well-nourished” (A), “suspected or moderately malnourished” (B), or “severely malnourished” (C), based on this scale.

### 2.6. Anthropometric Assessment

Nutritional status was evaluated through anthropometric measurements, encompassing weight (kg), height (cm), triceps skinfold thickness (mm), arm circumference (cm), and calf circumference (cm). Each measurement was conducted thrice, and the arithmetic mean was derived. Data collection occurred between July 2022 and December 2023, coinciding with the initiation of outpatient chemotherapy cycles for these patients.

### 2.7. Statistical Analysis

Categorical variables were presented as absolute and relative frequencies, while numerical variables were characterized using measures of central tendency and dispersion. The Wilcoxon test was employed to ascertain differences between subgroups of the PG-SGA score [22]. Associations among categorical variables were examined utilizing Fisher’s exact test [23]. Statistical analyses were performed using R software (version 4.3.2) and R Studio software (version 2023.09.1 build 494), with a predetermined significance level (alpha) set at 0.05.

## 3. Results

### Participants

Table 1 summarizes the sociodemographic, clinical, and dietary profiles of participants, revealing predominantly female (72.62%) residents from the metropolitan region of Espírito Santo (88.1%), with 61.9% identifying as non-white. About half (48.81%) attained primary education, with 76.1% reporting fixed incomes and 55% having partners. Clinically, 63.10% experienced delays exceeding 60 days before initiating chemotherapy, with Stage II being the most prevalent (57.14%).

Pre-existing conditions included systemic arterial hypertension (48%), diabetes mellitus (20%), and a history of COVID-19 (23%). A notable 66.6% had familial cancer history. Lifestyle habits indicated most participants were non-smokers (94.0%) and abstained from alcohol (86.9%). Most did not adhere to specific diets (89.2%), engage in physical activity (72.6%), reduce food intake (77.3%), seek nutritional counseling pre-diagnosis (84.5%) or post-diagnosis (70.2%), exhibit edema (88.1%), or frequently consume organic foods (51.9%).

Nutritional status, assessed via the PG-SGA, revealed 76 patients (90.48%) classified as well-nourished (59 female, 16 male) and 7 (8.33%) suspected of malnutrition, including 1 female and 6 males.

Table 2 presents the correlation between sociodemographic, clinical, and nutritional parameters and the PG-SGA classification of nutritional status. Among the variables examined, only gender exhibited a significant association with subjective global assessment, revealing a higher prevalence of well-nourished individuals among males. Conversely, males accounted for a greater proportion of patients with suspected malnutrition (six individuals) compared to females (one case).

Patients with cancer stages II and III demonstrated a higher incidence of suspected malnutrition compared to other stages. Notably, among the seven patients with suspected malnutrition, five had a family history of the disease.

Regarding nutritional counseling pre-diagnosis, a substantial proportion (71 individuals) reported no prior nutritional consultation, despite the majority (65 individuals) being classified as well-nourished. Conversely, among those with suspected malnutrition, a higher proportion lacked nutritional monitoring at this stage.

Table 3 compares means between the studied groups. It was observed that in the suspected malnutrition group, the average age was higher, whereas the average weight, BMI, waist circumference, arm circumference, and calf circumference were lower compared to the well-nourished group, with statistical significance.

## 4. Discussion

This study unveiled a 5.81% prevalence of suspected malnutrition among patients undergoing chemotherapy for non-metastatic cancer. Significant associations between sociodemographic, health, and clinical variables and nutritional status were observed, notably in age, weight, BMI, waist circumference, arm, and calf circumference.

Nutritional assessment assumes a pivotal role in patients undergoing multimodal antineoplastic treatment [13,24]. Addressing nutritional status from diagnosis onward and continual monitoring throughout treatment are imperative [19,24]. Malnutrition emerges as a common occurrence in these patients, exacerbating throughout anticancer interventions and detrimentally impacting quality of life [25].

The study highlights that cancer patients with lower body weight and BMI exhibit a heightened risk of malnutrition during the initial phases of chemotherapy infusion. The prevalence of malnutrition among cancer patients can vary widely, ranging from 30% to 70%, influenced by factors such as the screening tool utilized, cancer type, and treatment regimen [26]. For instance, in patients with head and neck cancer, nutritional deficits often manifest early in the disease course, with approximately 25% to 65% experiencing malnutrition characterized by weight loss exceeding 10% of normal body mass [27].

In addition to inadequate nutritional intake, metabolic disturbances, and stress, cancer patients frequently contend with chronic inflammation, further exacerbating weight loss and malnutrition. These factors significantly impact cancer progression, invasion, and metastasis [28].

Findings from a prospective multicenter cohort study by Latenstein et al. [29] focusing on pancreatic cancer align with our study, revealing that up to 71% of patients with this cancer subtype were malnourished at diagnosis, primarily due to the tumor’s effects. Notably, only 56% of these malnourished patients reported receiving nutritional intervention [29]. Another study evaluating cancer patients’ nutritional status using subjective global assessment demonstrated a direct correlation between malnutrition and poorer prognoses, as well as impairments in the patients’ quality of life [30].

In patients with esophageal cancer, malnutrition emerges as a predominant complication, often culminating in unfavorable prognoses and mortality. Comprehensive nutritional management not only enhances the nutritional status of these individuals but also mitigates the severity of radiation-induced esophagitis and skin reactions in patients undergoing concurrent chemotherapy and radiotherapy. Furthermore, it contributes to enhancing quality of life and alleviating depressive symptoms [31,32].

It is known that gastrointestinal cancers have a significant impact on the nutritional status of patients, and when combined with chemotherapy treatments, nutritional depletion becomes even more evident. Therefore, it is crucial that this process be as effective and tolerable as possible in order to minimize these severe stresses on the body [33].

Moreover, the study unveiled that the patients with smaller abdominal, arm, and calf circumference measurements exhibited a heightened risk of malnutrition compared to those with larger measurements. These findings resonate with the existing literature, underscoring the interplay between malnutrition and parameters such as weight loss, diminished muscle mass, and reduced anthropometric measurements in cancer patients. Additionally, our study reinforces the reliability of PG-SGA results in promptly identifying malnutrition in clinical settings [34,35].

Nutritional assessment, as per guideline recommendations, mandates the monitoring of nutritional deficiencies in all patients diagnosed with cancer. Malnutrition, irrespective of chemotherapy status, undermines functional capacity, compromises quality of life, heightens the risk of unplanned hospitalizations, and diminishes survival rates. Hence, targeted nutritional interventions tailored to individual needs are imperative for optimizing outcomes [36].

The study population revealed a propensity towards rapid weight loss and structural alterations such as decreased abdominal, arm, and calf circumference, potentially linked to diminished food intake and impaired functional capacity. Consequently, appropriate interventions must be tailored according to the severity of these manifestations. Patients’ dietary habits may be influenced by their cancer diagnosis, treatment regimen, and psychological and emotional factors. Inadequate nutrient intake can precipitate metabolic complications, exacerbating morbidity, mortality, and impeding treatment response [37].

Scientific evidence underscores the critical role of age in treatment costs, life expectancy, and overall health status, revealing a 1.5-fold higher morbidity and mortality rate for men compared to women across 29 cancer groups [38,39]. A survey encompassing 23,904 participants further demonstrated that the risk of nutritional deficiency in individuals with malignant cancer escalates progressively with age [40].

In our study, older participants exhibited a heightened risk of malnutrition. Additionally, despite there being mostly female participants, a notable difference in malnutrition risk was observed among male individuals. This finding resonates with De Groot et al.’s investigation involving 246 participants, averaging 61.9 years of age, wherein 31% were identified as at risk of malnutrition [41]. Moreover, corroborative evidence in the literature suggests that men tend to underutilize preventive health services and seek medical care later than women [42,43]. Consequently, factors such as health care stigma and perceptions of masculinity may significantly contribute to increased mortality risk.

Consistent with our findings, a retrospective cohort study involving 454 elderly cancer patients revealed that malnutrition correlated with an elevated risk of mortality. Specifically, researchers identified the male sex as a significant risk factor for mortality among these participants [44]. Early identification and ongoing monitoring of nutritional risk facilitate prompt detection and management of malnutrition [45,46], enabling patients to access timely nutritional support at the onset of cancer treatment. This proactive approach helps avert adverse prognoses associated with individual nutritional status.

In our study, most participants commenced treatment 60 days following diagnosis. The detrimental consequences of treatment delay were underscored by a systematic review and meta-analysis conducted by Hanna et al. [47], revealing that a four-week delay in treatment correlates with an elevated risk of mortality. This risk escalates further with delays extending to eight and twelve weeks [47]. The prolonged duration observed for most patients in our study may indicate systemic barriers within the healthcare system, such as limited access to specialized services or delays in scheduling appointments and procedures. Notably, delays in initiating treatment have been consistently linked to poorer prognoses and reduced survival rates [48,49].

While treatment delay may be attributed to factors such as the patient’s clinical stage or the necessity for additional diagnostic procedures or surgeries prior to chemotherapy initiation, emphasis is placed on minimizing delays to less than four weeks [46]. Surgical procedures play a key role in these patients, as they can be an additional factor contributing to nutritional depletion. Optimized recovery plays a crucial role after surgeries, especially in major procedures, as it significantly helps accelerate recovery and reduce complications [50].

While no association with the risk of malnutrition by PG-SGA was found, Stages II and III of cancer exhibited a greater prevalence of suspected malnutrition compared to other stages. This observation suggests a potential relationship between cancer stage and malnutrition risk, consistent with findings in the literature indicating that cancer location and stage are associated with patients’ nutritional risk [51,52].

Moreover, most patients in our study did not receive nutritional care before diagnosis, a trend that was more pronounced among those suspected of malnutrition. This finding resonates with previous studies highlighting the often-overlooked role of nutritional care in oncology treatment. Nutritional care has been demonstrated to be a crucial and predictive component of comprehensive assessment in oncology, emphasizing its importance in optimizing patient outcomes [53,54]. Additionally, several patients had a family history of cancer, underscoring the necessity for a multidisciplinary approach to enhance patient management.

### Study Limitations

This research has some limitations. Firstly, its cross-sectional methodological design precludes the establishment of the causal relationships investigated herein. Additionally, our sample size is small and derived from a single oncology reference center in Brazil, encompassing various malignant neoplasms in the analysis. Therefore, caution must be exercised when interpreting the data. Future research endeavors should employ well-designed studies with larger samples and longitudinal monitoring of patients, evaluating nutritional status at different intervals throughout chemotherapy treatment. Moreover, stratifying samples by specific cancer types is advisable for a more nuanced understanding.

Despite these limitations, our study has notable strengths. It focused on non-metastatic cancer patients (Stage I, II, and III) undergoing their first chemotherapy cycle, employing reliable methods for anthropometric measurements and nutritional status assessment endorsed by esteemed organizations such as the European Society for Medical Oncology, the European Society for Clinical Nutrition [25], and the Global Leadership Initiative in Malnutrition [54].

## 5. Conclusions

This study unveiled a significant prevalence of suspected malnutrition among non-metastatic cancer patients undergoing chemotherapy. Early nutritional screening and diagnosis are crucial for positively impacting patient health.

## Figures and Tables

**Table 1 nutrients-17-00168-t001:** Sociodemographic, clinical, and dietary characteristics of cancer patients undergoing chemotherapy (*n* = 84).

Variable	*n*	%
Sex		
Male	23	27.38
Female	61	72.62
Skin color		
White	32	38.10
Non-white	52	61.90
Educational attainment		
Illiterate	5	5.95
Elementary education	41	48.81
High school or higher education	38	45.24
Health regions of Espírito Santo		
Metropolitan region	74	88.10
Central-North and southern regions (non-metropolitan)	10	11.90
Income		
With income	64	76.19
Without income	20	23.81
Marital status		
With partner	47	55.95
Without partner	37	44.05
Time from diagnosis to start of treatment		
Up to 60 days	31	36.90
More than 60 days	53	63.10
Staging		
I	9	10.71
II	48	57.14
III	24	28.57
No information	3	3.57
Hypertension		
No	44	52.38
Yes	40	47.62
Diabetes mellitus		
No	68	80.95
Yes	16	19.05
COVID-19		
No	62	73.81
Yes	22	26.19
Family history of cancer		
No	28	33.33
Yes	56	66.67
Do you smoke?		
No	79	94.05
Yes	5	5.95
Do you consume alcohol?		
No	73	86.90
Yes	11	13.10
Do you follow any specific diet?		
No	75	89.29
Yes	9	10.71
Do you engage in any physical activity?		
No	61	72.62
Yes	23	27.38
Considering your typical food intake, do you feel that you have reduced your food consumption by half or less than half?		
No	65	77.38
Yes	19	22.62
Do you consume organic foods frequently?		
No	41	48.81
Yes	43	51.19
Were you already being followed by a nutritionist before the diagnosis?		
No	71	84.52
Yes	13	15.48
Did you receive care from a nutritionist after the diagnosis?		
No	59	70.24
Yes	25	29.76
Edema in limbs		
No	74	88.10
Yes	10	11.90

**Table 2 nutrients-17-00168-t002:** Nutritional assessment depends on sociodemographic, clinical, and nutritional variables.

Variables	Global Assessment		
	Well Nourished	Suspected Malnutrition	*p*-Value *
	*n* (%)	*n* (%)	
Sex			0.002
Male	17 (20.48)	6 (7.23)	
Female	59 (71.08)	1 (1.20)	
Skin color			0.707
White	29 (34.94)	2 (2.41)	
Non-white	47 (56.63)	5 (6.02)	
Education			1.000
Illiterate	5 (6.02)	0 (0.00)	
Elementary education	36 (43.37)	4 (4.82)	
High school or higher education	35 (42.17)	3 (3.61)	
Health regions of Espírito Santo			1.000
Metropolitan region	67 (80.72)	6 (7.23)	
Central-North and southern regions (non-metropolitan)	9 (10.84)	1 (1.20)	
Income			1.000
With income	57 (68.67)	6 (7.23)	
No income	19 (22.89)	1 (1.20)	
Marital status			0.132
With partner	41 (49.40)	6 (7.23)	
Without partner	35 (42.17)	1 (1.20)	
Time from diagnosis to start of treatment			0.416
Up to 60 days	27 (32.53)	4 (4.82)	
More than 60 days	49 (59.04)	3 (3.61)	
Staging			0.230
I	9 (10.84)	0 (0.00)	
II	44 (53.01)	3 (3.61)	
III	21 (25.30)	3 (3.61)	
No information	2 (2.41)	1 (1.20)	
Hypertension			1.000
No	40 (48.19)	4 (4.82)	
Yes	36 (43.37)	3 (3.61)	
Diabetes mellitus			1.000
No	62 (74.70)	6 (7.23)	
Yes	14 (16.87)	1 (1.20)	
COVID-19			1.000
No	56 (67.47)	5 (6.02)	
Yes	20 (24.10)	2 (2.41)	
Family history of cancer			1.000
No	26 (31.33)	2 (2.41)	
Yes	50 (60.24)	5 (6.02)	
Do you smoke?			0.364
No	72 (86.75)	6 (7.23)	
Yes	4 (4.82)	1 (1.20)	
Do you drink alcohol?			1.000
No	66 (79.52)	6 (7.23)	
Yes	10 (12.05)	1 (1.20)	
Do you follow any specific diet?			1.000
No	67 (80.72)	7 (8.43)	
Yes	9 (10.84)	0 (0.00)	
Do you engage in any physical activity?			0.667
No	54 (65.06)	6 (7.23)	
Yes	22 (26.51)	1 (1.20)	
Considering your typical food intake, do you feel that you have reduced your food consumption by half or less than half?			0.642
Yes	16 (19.28)	2 (2.41)	
No	60 (72.29)	5 (6.02)	
Do you consume organic foods?			0.713
Yes	39 (46.99)	3 (3.61)	
No	37 (44.58)	4 (4.82)	
Were you already being followed by a nutritionist before the diagnosis?			1.000
No	65 (78.31)	6 (7.23)	
Yes	11 (13.25)	1 (1.20)	
Did you receive care from a nutritionist after the diagnosis?			0.185
No	56 (67.47)	3 (3.61)	
Yes	20 (24.10)	4 (4.82)	
Edema in limbs			1.000
No	67 (80.72)	6 (7.23)	
Yes	9 (10.84)	1 (1.20)	
Weight loss (1 month)			0.866
0–1.9%	54 (65.06)	7 (8.43)	
2–2.9%	10 (12.05)	0 (0.00)	
3–4.9%	4 (4.82)	0 (0.00)	
5–9.9%	5 (6.02)	0 (0.00)	
10% or more	3 (3.61)	0 (0.00)	

* Fisher’s exact test.

**Table 3 nutrients-17-00168-t003:** Nutritional assessment depends on anthropometric and handgrip strength variables.

Variable	Global Assessment		
	Well-Nourished	Suspected Malnutrition	*p*-Value *
Age (years)	56.5 (11.69)	68.85 (5.90)	0.004
Number of children	2.37 (2.25)	2.86 (1.86)	0.298
Weight (kg)	74.10 (15.21)	58.39 (9.66)	0.005
BMI (kg/m²)	28.26 (5.50)	20.80 (1.59)	<0.001
Abdominal circumference (cm)	95.93 (12.51)	83.40 (8.96)	0.014
Waist circumference (cm)	90.78 (11.71)	81.90 (8.73)	0.058
Triceps skinfold thickness (mm)	29.83 (11.16)	20.86 (11.24)	0.057
Arm circumference (cm)	30.40 (4.92)	24.72 (2.37)	0.001
Calf circumference (cm)	36.66 (4.45)	32.75 (1.79)	0.009
Handgrip strength (mm)—Right	27.06 (16.85)	27.38 (11.73)	0.629
Handgrip strength (mm)—Left	29.38 (22.41)	24.86 (9.72)	0.9477
Blood glucose	126.31 (51.28)	104.71 (40.47)	0.136

* Wilcoxon–Mann–Whitney test.

## Data Availability

Data are contained within the article.

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
