# Peer review of "Nutritional Status of Patients with Neoplasms Undergoing Ambulatory Chemotherapy and Associated Factors"

_nutrients, 2025, doi:10.3390/nu17010168_

Round 1
Reviewer 1 Report
Comments and Suggestions for Authors
- I would suggest that the authors expand and reorganize the introduction to provide a more comprehensive and updated foundation for the study. Currently, the introduction does not fully capture the depth of research in this area, nor does it adequately contextualize the work within the broader scientific landscape. I recommend including a detailed overview of significant prior studies, which I have highlighted for your convenience, to ensure that the introduction both acknowledges and builds upon foundational research. This expanded context will enhance the reader’s understanding of the topic's relevance and the study’s contributions within the existing body of knowledge (the reviewer is not an Author of proposed studies): doi: 10.1097/JS9.0000000000000783; doi: 10.3389/fnut.2023.1045022
The methodology section would benefit from further clarification, particularly regarding the inclusion and exclusion criteria for the study. These criteria are not clearly reported, making it challenging to assess the selection process and the applicability of the findings. I recommend that the authors provide a more detailed description of the inclusion and exclusion criteria to enhance the rigor and reproducibility of the study.
The headings and descriptions of the tables in the manuscript should be more detailed to improve clarity and provide readers with a better understanding of the data presented. Clear and informative headings, along with comprehensive descriptions, are essential to convey the content and significance of each table without requiring extensive cross-referencing in the text. I recommend that the authors revise these elements to ensure that the tables are fully self-explanatory and effectively support the findings of the study.
The conclusions section would benefit from being rewritten in a clearer and more refined manner to align consistently with the results obtained in the study. Currently, the conclusions lack the precision and coherence needed to effectively convey the study's implications. I recommend that the authors revise this section to ensure that the conclusions are directly supported by the findings, offering a concise and logical summary of the study's contributions.
Comments on the Quality of English Language
The manuscript contains several weaknesses in English language usage, which impact the clarity and readability of the content. I recommend that the authors seek professional language editing to improve the grammar, syntax, and overall fluency, ensuring that the text meets the standards expected in academic writing.
Author Response
November 10th 2024
Dear Editor/Reviewer of Nutrients
We would like to express our gratitude for the valuable comments and suggestions provided to improve our manuscript. We carefully reviewed each recommendation and have implemented all the suggested changes.
Reviewer 1 :
- I would suggest that the authors expand and reorganize the introduction to provide a more comprehensive and updated foundation for the study. Currently, the introduction does not fully capture the depth of research in this area, nor does it adequately contextualize the work within the broader scientific landscape. I recommend including a detailed overview of significant prior studies, which I have highlighted for your convenience, to ensure that the introduction both acknowledges and builds upon foundational research. This expanded context will enhance the reader’s understanding of the topic's relevance and the study’s contributions within the existing body of knowledge (the reviewer is not an Author of proposed studies): doi: 10.1097/JS9.0000000000000783; doi: 10.3389/fnut.2023.1045022
Answer: Thank you very much. The introduction has been revised and made more direct to better contextualize the reader.
- The methodology section would benefit from further clarification, particularly regarding the inclusion and exclusion criteria for the study. These criteria are not clearly reported, making it challenging to assess the selection process and the applicability of the findings. I recommend that the authors provide a more detailed description of the inclusion and exclusion criteria to enhance the rigor and reproducibility of the study.
Answer: The methodology was adjusted and inclusion and exclusion criteria were added in detail.
- The headings and descriptions of the tables in the manuscript should be more detailed to improve clarity and provide readers with a better understanding of the data presented. Clear and informative headings, along with comprehensive descriptions, are essential to convey the content and significance of each table without requiring extensive cross-referencing in the text. I recommend that the authors revise these elements to ensure that the tables are fully self-explanatory and effectively support the findings of the study.
Answer: All tables have been revised, especially the titles and footnotes
- The conclusions section would benefit from being rewritten in a clearer and more refined manner to align consistently with the results obtained in the study. Currently, the conclusions lack the precision and coherence needed to effectively convey the study's implications. I recommend that the authors revise this section to ensure that the conclusions are directly supported by the findings, offering a concise and logical summary of the study's contributions.
Answer: The conclusion has been completely rewritten.
In addition, we have reduced the number of Author's or Nutrient's papers cited in the article as per recommended.
Reviewer 2 Report
Comments and Suggestions for Authors
Interesting paper that re-proposes a long-standing problem, the relationship between neoplasia and malnutrition. Good introduction that frames the general problem of malnutrition and the parameters for evaluating nutritional status. Over the years, nutrition doctors have taken into consideration a variety of data, from anthropometric to biohumoral indices, from which the Harris and Benedict formula emerged, allowing us to empirically calculate the nutritional needs of an individual based on the disease from which he was affected. During the 90s, nutritional therapy had its greatest development and it was possible to observe that all interventions had a better outcome if the patient arrived at the operating room in a good nutritional state. This is essentially the same concept presented by our colleagues. It is recommended, always in the introduction, to extract the diagnosis from their database to report in a table the main neoplasms they faced and which patients were most malnourished. This is because there is malnutrition linked to economic reasons and this is more or less endemic in all populations, but there is also malnutrition linked to neoplastic pathology, therefore esophageal or stomach cancer will lead the patient to be more malnourished than other forms of neoplasia. Furthermore, the paper rightly states that neoadjuvant therapy promotes loss of appetite and therefore malnutrition but there are works in the international literature that, as regards the stomach and esophagus affected by adenocarcinoma, can make use of the same neoadjuvant treatments but administered by another route, with less toxicity (DOI: 10.1097/CAD.0000000000000877 to be cited in the bibliography). We also agree that a bit all over the world there are centers to which malnourished patients can turn and it is true that it is often difficult to "make them loyal". We must not forget, however, that there are a series of products prepared by the industry that also enhance the immune status to improve the response to the surgical insult. Finally, one last aspect is suggested on which colleagues can spend a few words, the ERAS program that offers excellent results in the post-operative period (doi: 10.1177/0148607114523451 to be cited in the bibliography). We agree with the weak points, fair conclusion. Good iconography, English to be revised, good bibliography
Author Response
November 10th 2024
Dear Editor/Reviewer of Nutrients
We would like to express our gratitude for the valuable comments and suggestions provided to improve our manuscript. We carefully reviewed each recommendation and have implemented all the suggested changes.
Reviewer 2 :
- Interesting paper that re-proposes a long-standing problem, the relationship between neoplasia and malnutrition. Good introduction that frames the general problem of malnutrition and the parameters for evaluating nutritional status. Over the years, nutrition doctors have taken into consideration a variety of data, from anthropometric to biohumoral indices, from which the Harris and Benedict formula emerged, allowing us to empirically calculate the nutritional needs of an individual based on the disease from which he was affected. During the 90s, nutritional therapy had its greatest development and it was possible to observe that all interventions had a better outcome if the patient arrived at the operating room in a good nutritional state. This is essentially the same concept presented by our colleagues. It is recommended, always in the introduction, to extract the diagnosis from their database to report in a table the main neoplasms they faced and which patients were most malnourished. This is because there is malnutrition linked to economic reasons and this is more or less endemic in all populations, but there is also malnutrition linked to neoplastic pathology, therefore esophageal or stomach cancer will lead the patient to be more malnourished than other forms of neoplasia. Furthermore, the paper rightly states that neoadjuvant therapy promotes loss of appetite and therefore malnutrition but there are works in the international literature that, as regards the stomach and esophagus affected by adenocarcinoma, can make use of the same neoadjuvant treatments but administered by another route, with less toxicity (DOI: 10.1097/CAD.0000000000000877 to be cited in the bibliography). We also agree that a bit all over the world there are centers to which malnourished patients can turn and it is true that it is often difficult to "make them loyal". We must not forget, however, that there are a series of products prepared by the industry that also enhance the immune status to improve the response to the surgical insult. Finally, one last aspect is suggested on which colleagues can spend a few words, the ERAS program that offers excellent results in the post-operative period (doi: 10.1177/0148607114523451 to be cited in the bibliography). We agree with the weak points, fair conclusion. Good iconography, English to be revised, good bibliography
Answer: Thank you very much for your contributions and clear insights; we believe your expertise adds significant strength to our findings. The suggested articles have been added to the discussion.
The English was reviewed and edited for proper English by an English specialist with a doctorate degree and scientific experience in Health Sciences (certificate attached).
In addition, we have reduced the number of Author's or Nutrient's papers cited in the article as per recommended.
Thank you once again for your dedication and the detailed suggestions, which were essential in enhancing the quality of our work.
Round 2
Reviewer 1 Report
Comments and Suggestions for Authors
While the Author have clearly addressed all the previous raised issues, still remains to be addressed the following one:
-
- I would suggest that the authors expand and reorganize the introduction to provide a more comprehensive and updated foundation for the study. Currently, the introduction does not fully capture the depth of research in this area, nor does it adequately contextualize the work within the broader scientific landscape. I recommend including a detailed overview of significant prior studies, which I have highlighted for your convenience, to ensure that the introduction both acknowledges and builds upon foundational research. This expanded context will enhance the reader’s understanding of the topic's relevance and the study’s contributions within the existing body of knowledge (the reviewer is not an Author of proposed studies): doi: 10.1097/JS9.0000000000000783; doi: 10.3389/fnut.2023.1045022.
- For instance, the reference 5 is from 1985, please update it.
Author Response
November 14th 2024
Dear Editor/Reviewer of Nutrients
We would like to express our gratitude for the valuable comments and suggestions provided to improve our manuscript. We carefully reviewed each recommendation and have implemented all the suggested changes.
Reviewer 1 :
Comments and Suggestions for Authors
While the Author have clearly addressed all the previous raised issues, still remains to be addressed the following one:
- I would suggest that the authors expand and reorganize the introduction to provide a more comprehensive and updated foundation for the study. Currently, the introduction does not fully capture the depth of research in this area, nor does it adequately contextualize the work within the broader scientific landscape. I recommend including a detailed overview of significant prior studies, which I have highlighted for your convenience, to ensure that the introduction both acknowledges and builds upon foundational research. This expanded context will enhance the reader’s understanding of the topic's relevance and the study’s contributions within the existing body of knowledge (the reviewer is not an Author of proposed studies): doi: 10.1097/JS9.0000000000000783; doi: 10.3389/fnut.2023.1045022.
Answer: Ok. Done! Thank you for this suggestions. We have expanded the introduction context as well as added the references as per suggested.
- For instance, the reference 5 is from 1985, please update it.
Answer: Reference was updated as per recommended.
Thank you once again for your dedication and the detailed suggestions, which were essential in enhancing the quality of our work.